# Decomposable Submodular Function Minimization Discrete and Continuous

**Alina Ene**[*]  Huy L. Nguyễn[†]  László A. Végh[‡]

## Abstract

This paper investigates connections between discrete and continuous approaches for decomposable submodular function minimization. We provide improved running time estimates for the state-of-the-art continuous algorithms for the problem using combinatorial arguments. We also provide a systematic experimental comparison of the two types of methods, based on a clear distinction between level-0 and level-1 algorithms.

## 1 Introduction

Submodular functions arise in a wide range of applications: graph theory, optimization, economics, game theory, to name a few. A function $f : 2^V \to \mathbb{R}$ on a ground set $V$ is *submodular* if $f(X) + f(Y) \geq f(X \cap Y) + f(X \cup Y)$ for all sets $X, Y \subseteq V$. Submodularity can also be interpreted as a diminishing returns property.

There has been significant interest in submodular optimization in the machine learning and computer vision communities. The *submodular function minimization* (SFM) problem arises in problems in image segmentation or MAP inference tasks in Markov Random Fields. Landmark results in combinatorial optimization give polynomial-time exact algorithms for SFM. However, the high-degree polynomial dependence in the running time is prohibitive for large-scale problem instances. The main objective in this context is to develop fast and scalable SFM algorithms.

Instead of minimizing arbitrary submodular functions, several recent papers aim to exploit special structural properties of submodular functions arising in practical applications. This paper focuses on the popular model of *decomposable submodular functions*. These are functions that can be written as sums of several "simple" submodular functions defined on small supports.

Some definitions are needed to introduce our problem setting. Let $f : 2^V \to \mathbb{R}$ be a submodular function, and let $n := |V|$. We can assume w.l.o.g. that $f(\emptyset) = 0$. We are interested in solving the *submodular function minimization problem*:

$$\min_{S \subseteq V} f(S). \tag{SFM}$$

For a vector $y \in \mathbb{R}^V$ and a set $S \subseteq V$, we use the notation $y(S) := \sum_{v \in S} y(v)$. The base polytope of a submodular function is defined as

$$B(f) := \{y \in \mathbb{R}^V : y(S) \leq f(S) \ \forall S \subseteq V, y(V) = f(V)\}.$$

One can optimize linear functions over $B(f)$ using the greedy algorithm. The SFM problem can be reduced to finding the minimum norm point of the base polytope $B(f)$ [10].

$$\min \left\{ \frac{1}{2} \|y\|_2^2 \colon y \in B(f) \right\}. \tag{Min-Norm}$$

---

[*]Department of Computer Science, Boston University, `aene@bu.edu`

[†]College of Computer and Information Science, Northeastern University, `hu.nguyen@northeastern.edu`

[‡]Department of Mathematics, London School of Economics, `L.Vegh@lse.ac.uk`

This reduction is the starting point of convex optimization approaches for SFM. We refer the reader to Sections 44–45 in [28] for concepts and results in submodular optimization, and to [2] on machine learning applications.

We assume that $f$ is given in the decomposition $f(S) = \sum_{i=1}^{r} f_i(S)$, where each $f_i : 2^V \to \mathbb{R}$ is a submodular function. Such functions are called *decomposable* or *Sum-of-Submodular (SoS)* in the literature. In the *decomposable submodular function minimization (DSFM)* problem, we aim to minimize a function given in such a decomposition. We will make the following assumptions.

For each $i \in [r]$, we assume that two oracles are provided: *(i)* a value oracle that returns $f_i(S)$ for any set $S \subseteq V$ in time $\text{EO}_i$; and *(ii)* a quadratic minimization oracle $\mathcal{O}_i(w)$. For any input vector $w \in \mathbb{R}^n$, this oracle returns an optimal solution to (Min-Norm) for the function $f_i + w$, or equivalently, an optimal solution to $\min_{y \in B(f_i)} \|y + w\|_2^2$. We let $\Theta_i$ denote the running time of a single call to the oracle $\mathcal{O}_i$, $\Theta_{\max} := \max_{i \in [r]} \Theta_i$ denote the maximum time of an oracle call, $\Theta_{\text{avg}} := \frac{1}{r} \sum_{i \in [r]} \Theta_i$ denote the average time of an oracle call.[4] We let $F_{i,\max} := \max_{S \subseteq V} |f_i(S)|$, $F_{\max} := \max_{S \subseteq V} |f(S)|$ denote the maximum function values. For each $i \in [r]$, the function $f_i$ has an effective support $C_i$ such that $f_i(S) = f_i(S \cap C_i)$ for every $S \subseteq V$.

DSFM thus requires algorithms on two levels. The *level-0* algorithms are the subroutines used to evaluate the oracles $\mathcal{O}_i$ for every $i \in [r]$. The *level-1* algorithm minimizes the function $f$ using the level-0 algorithms as black boxes.

## 1.1 Prior work

SFM has had a long history in combinatorial optimization since the early 1970s, following the influential work of Edmonds [4]. The first polynomial-time algorithm was obtained via the ellipsoid method [14]; recent work presented substantial improvements using this approach [22]. Substantial work focused on designing strongly polynomial combinatorial algorithms [9, 15, 16, 25, 17, 27]. Still, designing practical algorithms for SFM that can be applied to large-scale problem instances remains an open problem.

Let us now turn to DSFM. Previous work mainly focused on level-1 algorithms. These can be classified as *discrete* and *continuous* optimization methods. The discrete approach builds on techniques of classical discrete algorithms for network flows and for submodular flows. Kolmogorov [21] showed that the problem can be reduced to submodular flow maximization, and also presented a more efficient augmenting path algorithm. Subsequent discrete approaches were given in [1, 7, 8]. Continuous approaches start with the convex programming formulation (Min-Norm). Gradient methods were applied for the decomposable setting in [5, 24, 30].

Less attention has been given to the level-0 algorithms. Some papers mainly focus on theoretical guarantees on the running time of level-1 algorithms, and treat the level-0 subroutines as black-boxes (e.g. [5, 24, 21]). In other papers (e.g. [18, 30]), the model is restricted to functions $f_i$ of a simple specific type that are easy to minimize. An alternative assumption is that all $C_i$'s are small, of size at most $k$; and thus these oracles can be evaluated by exhaustive search, in $2^k$ value oracle calls (e.g. [1, 7]). Shanu *et al.* [29] use a block coordinate descent method for level-1, and make no assumptions on the functions $f_i$. The oracles are evaluated via the Fujishige-Wolfe minimum norm point algorithm [11, 31] for level-0.

Let us note that these experimental studies considered the level-0 and level-1 algorithms as a single "package". For example, Shanu *et al.* [29] compare the performance of their *SoS Min-Norm algorithm* to the continuous approach of Jegelka *et al.* [18] and the combinatorial approach of Arora *et al.* [1]. However, these implementations cannot be directly compared, since they use three different level-0 algorithms: Fujishige-Wolfe in SoS Min-Norm, a general QP solver for the algorithm of [18], and exhaustive search for [1]. For potentials of large support, Fujishige-Wolfe outperforms these other level-0 subroutines, hence the level-1 algorithms in [18, 1] could have compared more favorably using the same Fujishige-Wolfe subroutine.

## 1.2 Our contributions

Our paper establishes connections between discrete and continuous methods for DSFM, as well as provides a systematic experimental comparison of these approaches. Our main theoretical contribution improves the worst-case complexity bound of the most recent continuous optimization methods [5, 24] by a factor of $r$, the number of functions in the decomposition. This is achieved by improving the bounds on the relevant condition numbers. Our proof exploits ideas from the discrete optimization approach. This provides not only better, but also considerably simpler arguments than the algebraic proof in [24].

The guiding principle of our experimental work is the clean conceptual distinction between the level-0 and level-1 algorithms, and to compare different level-1 algorithms by using the same level-0 subroutines. We compare the state-of-the-art continuous and discrete algorithms: RCDM and ACDM from [5] with Submodular IBFS from [7]. We consider multiple options for the level-0 subroutines. For certain potential types, we use tailored subroutines exploiting the specific form of the problem. We also consider a variant of the Fujishige-Wolfe algorithm as a subroutine applicable for arbitrary potentials.

Our experimental results reveal the following tradeoff. Discrete algorithms on level-1 require more calls to the level-0 oracle, but less overhead computation. Hence using algorithms such as IBFS on level-1 can be significantly faster than gradient descent, as long as the potentials have fairly small supports. However, as the size of the potentials grow, or we do need to work with a generic level-0 algorithm, gradient methods are preferable. Gradient methods can perform better for larger potentials also due to weaker requirements on the level-0 subroutines: approximate level-0 subroutines suffice for them, whereas discrete algorithms require exact optimal solutions on level-0.

**Paper outline.** The rest of the paper is structured as follows. The level-1 algorithmic frameworks using discrete and convex optimization are described in Sections 2 and 3, respectively. Section 4 gives improved convergence guarantees for the gradient descent algorithms outlined in Section 3. Section 5 discusses the different types of level-0 algorithms and how they can be used together with the level-1 frameworks. Section 6 presents a brief overview of our experimental results.

This is an extended abstract. The full paper is available on `http://arxiv.org/abs/1703.01830`.

## 2 Discrete optimization algorithms on Level-1

In this section, we outline a level-1 algorithmic framework for DSFM that is based on a combinatorial framework first studied by Fujishige and Zhang [12] for submodular intersection. The submodular intersection problem is equivalent to DSFM for the sum of two functions, and the approach can be adapted and extended to the general DSFM problem with an arbitrary decomposition. We now give a brief description of the algorithmic framework. The full version exhibits submodular versions of the Edmonds-Karp and preflow-push algorithms.

**Algorithmic framework.** For a decomposable function $f$, every $x \in B(f)$ can be written as $x = \sum_{i=1}^{r} x_i$, where $\mathrm{supp}(x_i) \subseteq C_i$ and $x_i \in B(f_i)$ (see e.g. Theorem 44.6 in [28]). A natural algorithmic approach is to maintain an $x \in B(f)$ in such a representation, and iteratively update it using the combinatorial framework described below. DSFM can be cast as a maximum network flow problem in a network that is suitably defined based on the current point $x$. This can be viewed as an analogue of the residual graph in the maxflow/mincut setting, and it is precisely the residual graph if the DSFM instance is a minimum cut instance.

**The auxiliary graph.** For an $x \in B(f)$ of the form $x = \sum_{i=1}^{r} x_i$, we construct the following directed auxiliary graph $G = (V, E)$, with $E = \bigcup_{i=1}^{r} E_i$ and capacities $c : E \to \mathbb{R}_+$. $E$ is a multiset union: we include parallel copies if the same arc occurs in multiple $E_i$. The arc sets $E_i$ are complete directed graphs (cliques) on $C_i$, and for an arc $(u, v) \in E_i$, we define $c(u, v) := \min\{f_i(S) - x_i(S): S \subseteq C_i, u \in S, v \notin S\}$. This is the maximum value $\varepsilon$ such that $x_i' \in B(f_i)$, where $x_i'(u) = x_i(u) + \varepsilon$, $x_i'(v) = x_i(v) - \varepsilon$, $x_i'(z) = x_i(z)$ for $z \notin \{u, v\}$.

Let $N := \{v \in V: x(v) < 0\}$ and $P := \{v \in V: x(v) > 0\}$. The algorithm aims to improve the current $x$ by updating along shortest directed paths from $N$ to $P$ with positive capacity; there are several ways to update the solution, and we discuss specific approaches (derived from maximum flow algorithms) in the full version. If there exists no such directed path, then we let $S$ denote the set

reachable from $N$ on directed paths with positive capacity; thus, $S \cap P = \emptyset$. One can show that $S$ is a minimizer of the function $f$.

Updating along a shortest path $\mathcal{Q}$ from $N$ to $P$ amounts to the following. Let $\varepsilon$ denote the minimum capacity of an arc on $\mathcal{Q}$. If $(u, v) \in \mathcal{Q} \cap E_i$, then we increase $x_i(u)$ by $\varepsilon$ and decrease $x_i(v)$ by $\varepsilon$. The crucial technical claim is the following. Let $d(u)$ denote the shortest path distance of positive capacity arcs from $u$ to the set $P$. Then, an update along a shortest directed path from $N$ to $P$ results in a feasible $x \in B(f)$, and further, all distance labels $d(u)$ are non-decreasing. We refer the reader to Fujishige and Zhang [12] for a proof of this claim.

**Level-1 algorithms based on the network flow approach.** Using the auxiliary graph described above, and updating on shortest augmenting paths, one can generalize several maximum flow algorithms to a level-1 algorithm of DSFM. In particular, based on the preflow-push algorithm [13], one can obtain a strongly polynomial DSFM algorithm with running time $O(n^2 \Theta_{\max} \sum_{i=1}^{r} |C_i|^2)$. A scaling variant provides a weakly polynomial running time $O(n^2 \Theta_{\max} \log F_{\max} + n \sum_{i=1}^{r} |C_i|^3 \Theta_i)$. We defer the details to the full version of the paper.

In our experiments, we use the submodular IBFS algorithm [7] as the main discrete level-1 algorithm; the same running time estimate as for preflow-push is applicable. If all $C_i$'s are small, $O(1)$, the running time is $O(n^2 r \Theta_{\max})$; note that $r = \Omega(n)$ in this case.

## 3 Convex optimization algorithms on Level-1

### 3.1 Convex formulations for DSFM

Recall the convex quadratic program (Min-Norm) from the Introduction. This program has a unique optimal solution $s^*$, and the set $S = \{v \in V : s^*(v) < 0\}$ is the unique smallest minimizer to the SFM problem. We will refer to this optimal solution $s^*$ throughout the section.

In the DSFM setting, one can write (Min-Norm) in multiple equivalent forms [18]. For the first formulation, we let $\mathcal{P} := \prod_{i=1}^{r} B(f_i) \subseteq \mathbb{R}^{rn}$, and let $A \in \mathbb{R}^{n \times (rn)}$ denote the following matrix:

$$A := \underbrace{[I_n I_n \ldots I_n]}_{r \text{ times}}.$$

Note that, for every $y \in \mathcal{P}$, $Ay = \sum_{i=1}^{r} y_i$, where $y_i$ is the $i$-th block of $y$, and thus $Ay \in B(f)$. The problem (Min-Norm) can be reformulated for DSFM as follows.

$$\min \left\{ \frac{1}{2} \|Ay\|_2^2 : y \in \mathcal{P} \right\}. \tag{Prox-DSFM}$$

The second formulation is the following. Let us define the subspace $\mathcal{A} := \{a \in \mathbb{R}^{nr} : Aa = 0\}$, and minimize its distance from $\mathcal{P}$:

$$\min \left\{ \|a - y\|_2^2 : a \in \mathcal{A}, y \in \mathcal{P} \right\}. \tag{Best-Approx}$$

The set of optimal solutions for both formulations (Prox-DSFM) and (Best-Approx) is the set $\mathcal{E} := \{y \in \mathcal{P} : Ay = s^*\}$, where $s^*$ is the optimum of (Min-Norm). We note that, even though the set of solutions to (Best-Approx) are pairs of points $(a, y) \in \mathcal{A} \times \mathcal{P}$, the optimal solutions are uniquely determined by $y \in \mathcal{P}$, since the corresponding $a$ is the projection of $y$ to $\mathcal{A}$.

### 3.2 Level-1 algorithms based on gradient descent

The gradient descent algorithms of [24, 5] provide level-1 algorithms for DSFM. We provide a brief overview of these algorithms and we refer the reader to the respective papers for more details.

**The alternating projections algorithm.** Nishihara *et al.* [24] minimize (Best-Approx) using *alternating projections*. The algorithm starts with a point $a_0 \in \mathcal{A}$ and it iteratively constructs a sequence $\{(a^{(k)}, x^{(k)})\}_{k \geq 0}$ by projecting onto $\mathcal{A}$ and $\mathcal{P}$: $x^{(k)} = \operatorname{argmin}_{x \in \mathcal{P}} \|a^{(k)} - x\|_2$, $a^{(k+1)} = \operatorname{argmin}_{a \in \mathcal{A}} \|a - x^{(k)}\|_2$.

**Random coordinate descent algorithms.** Ene and Nguyen [5] minimize (Prox-DSFM) using *random coordinate descent*. The RCDM algorithm adapts the random coordinate descent algorithm

of Nesterov [23] to (Prox-DSFM). In each iteration, the algorithm samples a block $i \in [r]$ uniformly at random and it updates $x_i$ via a standard gradient descent step for smooth functions. ACDM, the accelerated version of the algorithm, presents a further enhancement using techniques from [6].

## 3.3 Rates of convergence and condition numbers

The algorithms mentioned above enjoy a *linear convergence rate* despite the fact that the objective functions of (Best-Approx) and (Prox-DSFM) are not strongly convex. Instead, the works [24, 5] show that there are certain parameters that one can associate with the objective functions such that the convergence is at the rate $(1 - \alpha)^k$, where $\alpha \in (0, 1)$ is a quantity that depends on the appropriate parameter. Let us now define these parameters.

Let $\mathcal{A}'$ be the affine subspace $\mathcal{A}' := \{a \in \mathbb{R}^{nr} : Aa = s^*\}$. Note that the set $\mathcal{E}$ of optimal solutions to (Prox-DSFM) and (Best-Approx) is $\mathcal{E} = \mathcal{P} \cap \mathcal{A}'$. For $y \in \mathbb{R}^{nr}$ and a closed set $K \subseteq \mathbb{R}^{nr}$, we let $d(y, K) = \min \{\|y - z\|_2 : z \in K\}$ denote the distance between $y$ and $K$. The relevant parameter for the Alternating Projections algorithm is defined as follows.

**Definition 3.1** ([24]). For every $y \in (\mathcal{P} \cup \mathcal{A}') \setminus \mathcal{E}$, let

$$\kappa(y) := \frac{d(y, \mathcal{E})}{\max \{d(y, \mathcal{P}), d(y, \mathcal{A}')\}}, \quad \text{and} \quad \kappa_* := \sup \{\kappa(y) : y \in (\mathcal{P} \cup \mathcal{A}') \setminus \mathcal{E}\}.$$

The relevant parameter for the random coordinate descent algorithms is the following.

**Definition 3.2** ([5]). For every $y \in \mathcal{P}$, let $y^* := \mathrm{argmin}_p \{\|p - y\|_2 : p \in \mathcal{E}\}$ be the optimal solution to (Prox-DSFM) that is closest to $y$. We say that the objective function $\frac{1}{2}\|Ay\|_2^2$ of (Prox-DSFM) is *restricted $\ell$-strongly convex* if, for all $y \in \mathcal{P}$, we have

$$\|A(y - y^*)\|_2^2 \geq \ell \|y - y^*\|_2^2.$$

We define

$$\ell_* := \sup \left\{ \ell : \frac{1}{2}\|Ay\|_2^2 \text{ is restricted } \ell\text{-strongly convex} \right\}.$$

The running time dependence of the algorithms on these parameters is given in the following theorems.

**Theorem 3.3** ([24]). *Let* $(a^{(0)}, x^{(0)} = \mathrm{argmin}_{x \in \mathcal{P}} \|a^{(0)} - x\|_2)$ *be the initial solution and let* $(a^*, x^*)$ *be an optimal solution to (Best-Approx). The alternating projection algorithm produces in*

$$k = \Theta \left( \kappa_*^2 \ln \left( \frac{\|x^{(0)} - x^*\|_2}{\epsilon} \right) \right)$$

*iterations a pair of points* $a^{(k)} \in \mathcal{A}$ *and* $x^{(k)} \in \mathcal{P}$ *that is $\epsilon$-optimal, i.e.,*

$$\|a^{(k)} - x^{(k)}\|_2^2 \leq \|a^* - x^*\|_2^2 + \varepsilon.$$

**Theorem 3.4** ([5]). *Let* $x^{(0)} \in \mathcal{P}$ *be the initial solution and let* $x^*$ *be an optimal solution to (Prox-DSFM) that minimizes* $\|x^{(0)} - x^*\|_2$. *The random coordinate descent algorithm produces in*

$$k = \Theta \left( \frac{r}{\ell_*} \ln \left( \frac{\|x^{(0)} - x^*\|_2}{\epsilon} \right) \right)$$

*iterations a solution* $x^{(k)}$ *that is $\epsilon$-optimal in expectation, i.e.,* $\mathbb{E} \left[ \frac{1}{2}\|Ax^{(k)}\|_2^2 \right] \leq \frac{1}{2}\|Ax^*\|_2^2 + \epsilon$.

*The accelerated coordinate descent algorithm produces in*

$$k = \Theta \left( r \sqrt{\frac{1}{\ell_*}} \ln \left( \frac{\|x^{(0)} - x^*\|_2}{\epsilon} \right) \right)$$

*iterations (specifically,* $\Theta \left( \ln \left( \frac{\|x^{(0)} - x^*\|_2}{\epsilon} \right) \right)$ *epochs with* $\Theta \left( r \sqrt{\frac{1}{\ell_*}} \right)$ *iterations in each epoch) a solution* $x^{(k)}$ *that is $\epsilon$-optimal in expectation, i.e.,* $\mathbb{E} \left[ \frac{1}{2}\|Ax^{(k)}\|_2^2 \right] \leq \frac{1}{2}\|Ax^*\|_2^2 + \epsilon$.

## 3.4 Tight analysis for the condition numbers and running times

We provide a tight analysis for the condition numbers (the parameters $\kappa_*$ and $\ell_*$ defined above). This leads to improved upper bounds on the running times of the gradient descent algorithms.

**Theorem 3.5.** *Let $\kappa_*$ and $\ell_*$ be the parameters defined in Definition 3.1 and Definition 3.2. We have $\kappa_* = \Theta(n\sqrt{r})$ and $\ell_* = \Theta(1/n^2)$.*

Using our improved convergence guarantees, we obtain the following improved running time analyses.

**Corollary 3.6.** *The total running time for obtaining an $\epsilon$-approximate solution[5] is as follows.*

- *Alternating projections (AP): $O\left(n^2 r^2 \Theta_{\mathrm{avg}} \ln \left( \frac{\|x^{(0)} - x^*\|_2}{\epsilon} \right)\right).$*

- *Random coordinate descent (RCDM): $O\left(n^2 r \Theta_{\mathrm{avg}} \ln \left( \frac{\|x^{(0)} - x^*\|_2}{\epsilon} \right)\right).$*

- *Accelerated random coordinate descent (ACDM): $O\left(n r \Theta_{\mathrm{avg}} \ln \left( \frac{\|x^{(0)} - x^*\|_2}{\epsilon} \right)\right).$*

We can upper bound the diameter of the base polytope by $O(\sqrt{n}F_{\max})$ [19], and thus $\|x^{(0)} - x^*\|_2 = O(\sqrt{n}F_{\max})$. For integer-valued functions, a $\varepsilon$-approximate solution can be converted to an exact optimum if $\varepsilon = O(1/n)$ [2].

The upper bound on $\kappa_*$ and the lower bound on $\ell_*$ are shown in Theorem 4.2. The lower bound on $\kappa_*$ and upper bound on $\ell_*$ in Theorem 3.5 follow by constructions in previous work, as explained next. Nishihara *et al.* showed that $\kappa_* \le nr$, and they give a family of minimum cut instances for which $\kappa_* = \Omega(n\sqrt{r})$. Namely, consider a graph with $n$ vertices and $m$ edges, and suppose for simplicity that the edges have integer capacities at most $C$. The cut function of the graph can be decomposed into functions corresponding to the individual edges, and thus $r = m$ and $\Theta_{\mathrm{avg}} = O(1)$. Already on simple cycle graphs, they show that the running time of AP is $\Omega(n^2 m^2 \ln(nC))$, which implies $\kappa_* = \Omega(n\sqrt{r})$.

Using the same construction, it is easy to obtain the upper bound $\ell_* = O(1/n^2)$.

## 4 Tight convergence bounds for the convex optimization algorithms

In this section, we show that the combinatorial approach introduced in Section 2 can be applied to obtain better bounds on the parameters $\kappa_*$ and $\ell_*$ defined in Section 3. Besides giving a stronger bound, our proof is considerably simpler than the algebraic one using Cheeger's inequality in [24]. The key is the following lemma.

**Lemma 4.1.** *Let $y \in \mathcal{P}$ and $s^* \in B(f)$. Then there exists a point $x \in \mathcal{P}$ such that $Ax = s^*$ and $\|x - y\|_2 \le \frac{\sqrt{n}}{2}\|Ay - s^*\|_1$.*

Before proving this lemma, we show how it can be used to derive the bounds.

**Theorem 4.2.** *We have $\kappa_* \le n\sqrt{r}/2 + 1$ and $\ell_* \ge 4/n^2$.*

**Proof:** We start with the bound on $\kappa_*$. In order to bound $\kappa_*$, we need to upper bound $\kappa(y)$ for any $y \in (\mathcal{P} \cup \mathcal{A}') \setminus \mathcal{E}$. We distinguish between two cases: $y \in \mathcal{P} \setminus \mathcal{E}$ and $y \in \mathcal{A}' \setminus \mathcal{E}$.

**Case I: $y \in \mathcal{P} \setminus \mathcal{E}$.** The denominator in the definition of $\kappa(y)$ is equal to $d(y, \mathcal{A}') = \|Ay - s^*\|_2 / \sqrt{r}$. This follows since the closest point $a = (a_1, \ldots, a_r)$ to $y$ in $\mathcal{A}'$ can be obtained as $a_i = y_i + (s^* - Ay)/r$ for each $i \in [r]$. Lemma 4.1 gives an $x \in \mathcal{P}$ such that $Ax = s^*$ and $\|x - y\|_2 \le \frac{\sqrt{n}}{2}\|Ay - s^*\|_1 \le \frac{n}{2}\|Ay - s^*\|_2$. Since $Ax = s^*$, we have $x \in \mathcal{E}$ and thus the numerator of $\kappa(y)$ is at most $\|x - y\|_2$. Thus $\kappa(y) \le \|x - y\|_2 / (\|Ay - s^*\|_2 / \sqrt{r}) \le n\sqrt{r}/2$.

**Case II: $y \in \mathcal{A}' \setminus \mathcal{E}$.** This means that $Ay = s^*$. The denominator of $\kappa(y)$ is equal to $d(y, \mathcal{P})$. For each $i \in [r]$, let $q_i \in B(f_i)$ be the point that minimizes $\|y_i - q_i\|_2$. Let $q = (q_1, \ldots, q_r) \in \mathcal{P}$. Then

$d(y, \mathcal{P}) = \|y - q\|_2$. Lemma 4.1 with $q$ in place of $y$ gives a point $x \in \mathcal{E}$ such that $\|q - x\|_2 \leq \frac{\sqrt{n}}{2}\|Aq - s^*\|_1$. We have $\|Aq - s^*\|_1 = \|Aq - Ay\|_1 \leq \sum_{i=1}^r \|q_i - y_i\|_1 = \|q - y\|_1 \leq \sqrt{nr}\|q - y\|_2$. Thus $\|q - x\|_2 \leq \frac{n\sqrt{r}}{2}\|q - y\|_2$. Since $x \in \mathcal{E}$, we have $d(y, \mathcal{E}) \leq \|x - y\|_2 \leq \|x - q\|_2 + \|q - y\|_2 \leq \left(1 + \frac{n\sqrt{r}}{2}\right)\|q - y\|_2 = \left(1 + \frac{n\sqrt{r}}{2}\right) d(y, \mathcal{P})$. Therefore $\kappa(p) \leq 1 + \frac{n\sqrt{r}}{2}$, as desired.

Let us now prove the bound on $\ell_*$. Let $y \in \mathcal{P}$ and let $y^* := \mathrm{argmin}_p\{\|p - y\|_2 : y \in \mathcal{E}\}$. We need to verify that $\|A(y - y^*)\|_2^2 \geq \frac{4}{n^2}\|y - y^*\|_2^2$. Again, we apply Lemma 4.1 to obtain a point $x \in \mathcal{P}$ such that $Ax = s^*$ and $\|x - y\|_2^2 \leq \frac{n}{4}\|Ax - Ay\|_1^2 \leq \frac{n^2}{4}\|Ax - Ay\|_2^2$. Since $Ax = s^*$, the definition of $y^*$ gives $\|y - y^*\|_2^2 \leq \|x - y\|_2^2$. Using that $Ax = Ay^* = s^*$, we have $\|Ax - Ay\|_2 = \|Ay - Ay^*\|_2$. $\qquad\square$

**Proof of Lemma 4.1:** We give an algorithm that transforms $y$ to a vector $x \in \mathcal{P}$ as in the statement through a sequence of path augmentations in the auxiliary graph defined in Section 2. We initialize $x = y$ and maintain $x \in \mathcal{P}$ (and thus $Ax \in B(f)$) throughout. We now define the set of source and sink nodes as $N := \{v \in V : (Ax)(v) < s^*(v)\}$ and $P := \{v \in V : (Ax)(v) > s^*(v)\}$. Once $N = P = \emptyset$, we have $Ax = s^*$ and terminate. Note that since $Ax, s^* \in B(f)$, we have $\sum_v (Ax)(v) = \sum_v s^*(v) = f(V)$, and therefore $N = \emptyset$ is equivalent to $P = \emptyset$. The blocks of $x$ are denoted as $x = (x_1, x_2, \dots, x_r)$, with $x_i \in B(f_i)$.

**Claim 4.3.** *If $N \neq \emptyset$, then there exists a directed path of positive capacity in the auxiliary graph between the sets $N$ and $P$.*

**Proof:** We say that a set $T$ is $i$-tight, if $x_i(T) = f_i(T)$. It is a simple consequence of submodularity that the intersection and union of two $i$-tight sets are also $i$-tight sets. For every $i \in [r]$ and every $u \in V$, we define $T_i(u)$ as the unique minimal $i$-tight set containing $u$. It is easy to see that for an arc $(u, v) \in E_i$, $c(u, v) > 0$ if and only if $v \in T_i(u)$. We note that if $u \notin C_i$, then $x(u) = f_i(\{u\}) = 0$ and thus $T_i(u) = \{u\}$.

Let $S$ be the set of vertices reachable from $N$ on a directed path of positive capacity in the auxiliary graph. For a contradiction, assume $S \cap P = \emptyset$. By the definition of $S$, we must have $T_i(u) \subseteq S$ for every $u \in S$ and every $i \in [r]$. Since the union of $i$-tight sets is also $i$-tight, we see that $S$ is $i$-tight for every $i \in [r]$, and consequently, $x(S) = f(S)$. On the other hand, since $N \subseteq S$, $S \cap P = \emptyset$, and $N \neq \emptyset$, we have $x(S) < s^*(S)$. Since $s^* \in B(f)$, we have $f(S) = x(S) < s^*(S) \leq f(S)$, a contradiction. We conclude that $S \cap P \neq \emptyset$. $\qquad\square$

In every step of the algorithm, we take a shortest directed path $\mathcal{Q}$ of positive capacity from $N$ to $P$, and update $x$ along this path. That is, if $(u, v) \in \mathcal{Q} \cap E_i$, then we increase $x_i(u)$ by $\varepsilon$ and decrease $x_i(v)$ by $\varepsilon$, where $\varepsilon$ is the minimum capacity of an arc on $\mathcal{Q}$. Note that this is the same as running the Edmonds-Karp-Dinitz algorithm in the submodular auxiliary graph. Using the analysis of [12], one can show that this change maintains $x \in \mathcal{P}$, and that the algorithm terminates in finite (in fact, strongly polynomial) time. We defer the details to the full version of the paper.

It remains to bound $\|x - y\|_2$. At every path update, the change in $\ell_\infty$-norm of $x$ is at most $\varepsilon$, and the change in $\ell_1$-norm is at most $n\varepsilon$, since the length of the path is $\leq n$. At the same time, $\sum_{v \in N}(s^*(v) - (Ax)(v))$ decreases by $\varepsilon$. Thus, $\|x - y\|_\infty \leq \|Ay - s^*\|_1/2$ and $\|x - y\|_1 \leq n\|Ay - s^*\|_1/2$. Using the inequality $\|p\|_2 \leq \sqrt{\|p\|_1\|p\|_\infty}$, we obtain $\|x - y\|_2 \leq \frac{\sqrt{n}}{2}\|Ay - s^*\|_1$, completing the proof. $\qquad\square$

## 5 The level-0 algorithms

In this section, we briefly discuss the level-0 algorithms and the interface between the level-1 and level-0 algorithms.

**Two-level frameworks via quadratic minimization oracles.** Recall from the Introduction the assumption on the subroutines $\mathcal{O}_i(w)$ that finds the minimum norm point in $B(f_i + w)$ for the input vector $w \in \mathbb{R}^n$ for each $i \in [r]$. The continuous methods in Section 3 directly use the subroutines $\mathcal{O}_i(w)$ for the alternating projection or coordinate descent steps. For the flow-based algorithms in Section 2, the main oracle query is to find the auxiliary graph capacity $c(u, v)$ of an arc $(u, v) \in E_i$ for some $i \in [r]$. This can be easily formulated as minimizing the function $f_i + w$ for an appropriate $w$ with $\mathrm{supp}(w) \subseteq C_i$. As explained at the beginning of Section 3, an optimal solution to (Min-Norm)

immediately gives an optimal solution to the SFM problem for the same submodular function. Hence, the auxiliary graph capacity queries can be implemented via single calls to the subroutines $\mathcal{O}_i(w)$. Let us also remark that, while the functions $f_i$ are formally defined on the entire ground set $V$, their effective support is $C_i$, and thus it suffices to solve the quadratic minimization problems on the ground set $C_i$.

Whereas discrete and continuous algorithms require the same type of oracles, there is an important difference between the two algorithms in terms of exactness for the oracle solutions. The discrete algorithms require exact values of the auxiliary graph capacities $c(u, v)$, as they must maintain $x_i \in B(f_i)$ throughout. Thus, the oracle must always return an optimal solution. The continuous algorithms are more robust, and return a solution with the required accuracy even if the oracle only returns an approximate solution. As discussed in Section 6, this difference leads to the continuous methods being applicable in settings where the combinatorial algorithms are prohibitively slow.

**Level-0 algorithms.** We now discuss specific algorithms for quadratic minimization over the base polytopes of the functions $f_i$. Several functions that arise in applications are "simple", meaning that there is a function-specific quadratic minimization subroutine that is very efficient. If a function-specific subroutine is not available, one can use a general-purpose submodular minimization algorithm. The works [1, 7] use a *brute force search* as the subroutine for each each $f_i$, whose running time is $2^{|C_i|}\mathrm{EO}_i$. However, this is applicable only for small $C_i$'s and is not suitable for our experiments where the maximum clique size is quite large. As a general-purpose algorithm, we used the *Fujishige-Wolfe minimum norm point algorithm* [11, 31]. This provides an $\varepsilon$-approximate solution in $O(|C_i|F_{i,\max}^2/\varepsilon)$ iterations, with overall running time bound $O((|C_i|^4 + |C_i|^2\mathrm{EO}_i)F_{i,\max}^2/\varepsilon)$ [3]. The experimental running time of the Fujishige-Wolfe algorithm can be prohibitively large [20]. As we discuss in Section 6, by warm-starting the algorithm and performing only a small number of iterations, we were able to use the algorithm in conjunction with the gradient descent level-1 algorithms.

## 6 Experimental results

We evaluate the algorithms on energy minimization problems that arise in image segmentation problems. We follow the standard approach and model the image segmentation task of segmenting an object from the background as finding a minimum cost $0/1$ labeling of the pixels. The total labeling cost is the sum of labeling costs corresponding to *cliques*, where a clique is a set of pixels. We refer to the labeling cost functions as clique potentials.

The main focus of our experimental analysis is to compare the running times of the decomposable submodular minimization algorithms. Therefore we have chosen to use the simple hand-tuned potentials that were used in previous work: the *edge-based costs* [1] and the *count-based costs* defined by [29, 30]. Specifically, we used the following clique potentials in our experiments, all of which are submodular:

- **Unary potentials** for each pixel. The unary potentials are derived from Gaussian Mixture Models of color features [26].
- **Pairwise potentials** for each edge of the 8-neighbor grid graph. For each graph edge $(i, j)$ between pixels $i$ and $j$, the cost of a labeling equals 0 if the two pixels have the same label, and $\exp(-\|v_i - v_j\|^2)$ for different labels, where $v_i$ is the RGB color vector of pixel $i$.
- **Square potentials** for each $2 \times 2$ square of pixels. The cost of a labeling is the square root of the number of neighboring pixels that have different labels, as in [1].
- **Region potentials.** We use the algorithm from [30] to identify regions. For each region $C_i$, the labeling cost is $f_i(S) = |S||C_i \setminus S|$, where $S$ and $C_i \setminus S$ are the subsets of $C_i$ labeled 0 and 1, respectively, see [29, 30].

We used five image segmentation instances to evaluate the algorithms.[6] The experiments were carried out on a single computer with a 3.3 GHz Intel Core i5 processor and 8 GB of memory; we reported averaged times over 10 trials.

We performed several experiments with various combinations of potentials and parameters. In the *minimum cut experiments*, we evaluated the algorithms on instances containing only unary and

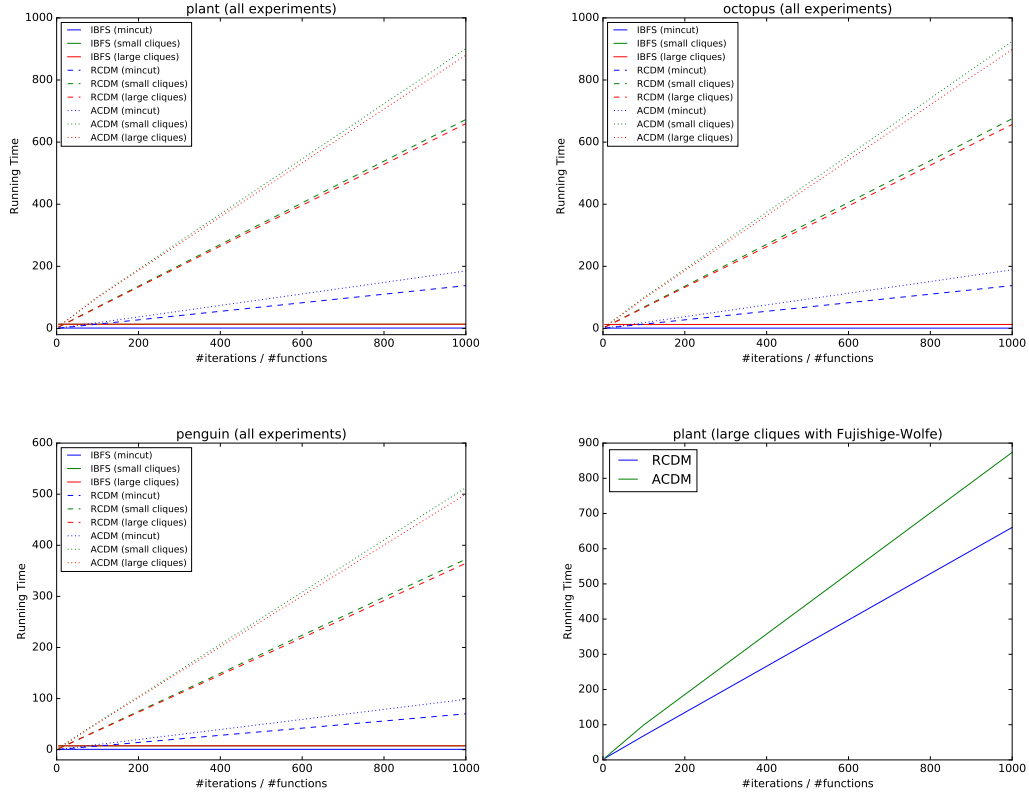

Figure 1: Running times in seconds on a subset of the instances. The results for the other instances are very similar and are deferred to the full version of the paper. The $x$-axis shows the number of iterations for the continuous algorithms. The IBFS algorithm is exact, and we display its running time as a flat line. In the first three plots, the running time of IBFS on the small cliques instances nearly coincides with its running time on minimum cut instances. In the last plot, the running time of IBFS is missing since it is computationally prohibitive to run it on those instances.

pairwise potentials; in the *small cliques experiments*, we used unary, pairwise, and square potentials. Finally, the *large cliques experiments* used all potentials above. Here, we used two different level-0 algorithms for the region potentials. Firstly, we used an algorithm specific to the particular potential, with running time $O(|C_i| \log(|C_i|) + |C_i| \mathrm{EO}_i)$. Secondly, we used the general Fujishige-Wolfe algorithm for level-0. This turned out to be significantly slower: it was prohibitive to run the algorithm to near-convergence. Hence, we could not implement IBFS in this setting as it requires an exact solution.

We were able to implement coordinate descent methods with the following modification of Fujishige-Wolfe at level-0. At every iteration, we ran Fujishige-Wolfe for 10 iterations only, but we *warm-started* with the current solution $x_i \in B(f_i)$ for each $i \in [r]$. Interestingly, this turned out to be sufficient for the level-1 algorithm to make progress.

**Summary of results.** Figure 1 shows the running times for some of the instances; we defer the full experimental results to the full version of the paper. The IBFS algorithm is significantly faster than the gradient descent algorithms on all of the instances with small cliques. For all of the instances with larger cliques, IBFS (as well as other combinatorial algorithms) are no longer suitable if the only choice for the level-0 algorithms are generic methods such as the Fujishige-Wolfe algorithm. The experimental results suggest that in such cases, the coordinate descent methods together with a suitably modified Fujishige-Wolfe algorithm provides an approach for obtaining an approximate solution.

## Footnotes

[4]For flow-type algorithms for DSFM, a slightly weaker oracle assumption suffices, returning a minimizer of $\min_{S \subseteq C_i} f_i(S) + w(S)$ for any given $w \in \mathbb{R}^{C_i}$. This oracle and the quadratic minimization oracle are reducible to each other: the former reduces to a single call to the latter, and one can implement the latter using $O(|C_i|)$ calls to the former (see e.g. [2]).

[5] The algorithms considered here solve the optimization problem (Prox-DSFM). An $\varepsilon$-approximate solution to an optimization problem $\min\{f(x) : x \in P\}$ is a solution $x \in P$ satisfying $f(x) \le f(x^*) + \varepsilon$, where $x^* \in \operatorname{argmin}_{x \in P} f(x)$ is an optimal solution.

[6]The data is available at `http://melodi.ee.washington.edu/~jegelka/cc/index.html` and `http://research.microsoft.com/en-us/um/cambridge/projects/visionimagevideoediting/segmentation/grabcut.htm`

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
