[Reviews · NeurIPS 2017]

Reviewer 1



This paper studies the problem of minimizing a decomposable submodular function. Submodular minimization is a well studied and important problem in machine learning for which there exist algorithms to solve the problem exactly. However, the running time of these algorithms is a high polynomial and they are thus oftentimes not practical. To get around this issue, submodular functions that can be decomposed and written as a sum of submodular functions over a much smaller support (DSFM) are often considered as they often appear in practice. This paper improves the analysis of the fastest algorithms for DSFM by a factor equal to the number of functions in the decomposition. It also provides an experimental framework based on a distinction between “level 0” algorithms, which are subroutines for quadratic minimization, and “level 1” algorithms which minimize the function using level 0 as a black box. These allow a more meaningful comparison where the same level 0 algorithms are used to compare different algorithms. These experiments show a tradeoff between the discrete algorithms that require more calls to the level 0 subroutines and gradient methods with weaker requirements for level 0 but more computation for level 1. The analysis is complex and relies on both discrete and continuous optimization techniques to meaningfully improve the running time of an important problem where the computational complexity is expensive. The experiments also highlight an interesting tradeoff which suggests that different algorithms should be used in different contexts for the running time of DSFM. A weakness of the paper is that the writing is very dense and sometimes hard to follow. It would have been nice to have more discussion on the parameters kappa* and l* and the precise bounds in terms of these parameters. It would have also been nice to have some comparison on the theoretical bounds between RCDM, ACDM, and IBFS.

Reviewer 2



This paper investigates connections between discrete and continuous approaches for decomposable submodular function minimization. The authors claim better theoretical time complexity bounds and experimental results on image segmentation. 1.In line 73 of page 2, instead of saying “improves the worst-case complexity by a factor of r”, it is better to clarify what the current state-of-art result is, what is the paper’s contribution; otherwise, readers won’t know the significant of your results. 2. In line 140 of page, the bounds contain the maximum value of the function. But this only makes sense when the function is integer-valued. The author should be clearer on this. Also, are these algorithms weakly polynomial or strongly polynomial? 3. In line 144, it is better to notify the reader that when C_i is small ~O(1), r has to be no smaller than ~O(n). Also, how do you compare your results with the state-of-art submodular minimization running time? 4. In Corollary 3.4, what is the definition of epsilon-approximate? 5 When introducing the algorithms, it is better to use \algorithm environments than using plain text.